# Comparative Proteomics Analysis between Maize and Sorghum Uncovers Important Proteins and Metabolic Pathways Mediating Drought Tolerance

**DOI:** 10.3390/life13010170

**Published:** 2023-01-06

**Authors:** Ali Elnaeim Elbasheir Ali, Lizex Hollenbach Husselmann, David L. Tabb, Ndiko Ludidi

**Affiliations:** 1Department of Biotechnology, University of the Western Cape, Robert Sobukwe Road, Bellville 7530, South Africa; 2Centre for Bioinformatics and Computational Biology, Division of Molecular Biology and Human Genetics, Faculty of Medicine and Health Sciences, Stellenbosch University, Cape Town 7500, South Africa; 3DSI-NRF Centre of Excellence in Food Security, University of the Western Cape, Robert Sobukwe Road, Bellville 7530, South Africa

**Keywords:** proteomic analysis, drought stress, maize, sorghum, drought tolerance

## Abstract

Drought severely affects crop yield and yield stability. Maize and sorghum are major crops in Africa and globally, and both are negatively impacted by drought. However, sorghum has a better ability to withstand drought than maize. Consequently, this study identifies differences between maize and sorghum grown in water deficit conditions, and identifies proteins associated with drought tolerance in these plant species. Leaf relative water content and proline content were measured, and label-free proteomics analysis was carried out to identify differences in protein expression in the two species in response to water deficit. Water deficit enhanced the proline accumulation in sorghum roots to a higher degree than in maize, and this higher accumulation was associated with enhanced water retention in sorghum. Proteomic analyses identified proteins with differing expression patterns between the two species, revealing key metabolic pathways that explain the better drought tolerance of sorghum than maize. These proteins include phenylalanine/tyrosine ammonia-lyases, indole-3-acetaldehyde oxidase, sucrose synthase and phenol/catechol oxidase. This study highlights the importance of phenylpropanoids, sucrose, melanin-related metabolites and indole acetic acid (auxin) as determinants of the differences in drought stress tolerance between maize and sorghum. The selection of maize and sorghum genotypes with enhanced expression of the genes encoding these differentially expressed proteins, or genetically engineering maize and sorghum to increase the expression of such genes, can be used as strategies for the production of maize and sorghum varieties with improved drought tolerance.

## 1. Introduction

*Sorghum bicolor*, commonly known as sorghum, and *Zea mays*, commonly called maize, are two of the major staple cereals in the world belonging to the Panicoideae subfamily in the family Gramineae [1]. Under drought conditions, sustained crop production is necessary to ensure global food security and requires accelerated crop breeding to develop drought-tolerant crops [2]. Sorghum is a candidate for this breeding effort due to its adaptation to drought [3]. The completed sequencing of the sorghum genome makes it a key model system for understanding the drought-responsive molecular mechanisms in plants [4]. Sorghum ranks as the fifth most significant crop across the globe after maize, rice, wheat and barley [5,6]. It is cultivated for food, feed and biofuel production. In Africa, sorghum is the second major grain after maize, with an annual production of approximately 20 million tons in the continent, which contributes one-third of the global crop production [6]. Globally, maize is the third main cereal in terms of harvested area [7] and serves as a staple in sub-Saharan Africa [8]. Maize is the main grain crop grown in South Africa, with approximately 12.2 million tons produced annually in the country [9]. It also acts as a source of biofuel and starch.

Drought is considered as one of the most significant natural hazards, and its intensity and frequency are projected to increase due to global warming [10]. According to recent reports, drought has affected 2.3 billion people across the world, with African communities affected the most as the continent accounts for 40% of the world total [11,12]. Based on annual rain fall data, South Africa is one of the 30 driest countries in the world [13]. In the last four decades, drought has become more prevalent in the country, negatively affecting agriculture and magnifying food insecurity in the region [14]. Among the major threats to crop production, drought is the most significant [15]. It has been reported that over the period from 1983 to 2009, three-quarters of the cultivated areas of key crops in the world, including maize, rice, soybean and wheat, experienced yield losses due to droughts [16]. The crop yield losses per drought event during that period were 7% for maize and soybean, 8% for wheat and 3% for rice. Improving sustainable crop production under conditions of limited water supply is important to meet the increasing food demand of the world’s growing population [17]. In arid or semi-arid regions, such as South Africa, screening of the adaptive responses to drought in crops is essential to the improvement of crop production under water deficit. Drought tolerance indices can be obtained by assessing the molecular responses to water deficit in crop plants, providing more insight into the mechanisms that may improve drought tolerance in the plants.

Recent advances in proteomic approaches have significantly improved the identification of a wide range of proteins in living cells [18]. This aspect is particularly important and useful for crop science. This is because it may augment the understanding of the molecular mechanisms that regulate the processes involved in the determination of the yield and nutrient content in crops. Advances in proteomics will help in elucidating how the yield and nutrient content are affected by adverse conditions such as stress resulting from drought [19]. Proteomics, one of the key tools of the post-genomic era [20], offers sensitive identification of the proteins associated with drought responses in plants [21,22,23,24].

Although several studies have already reported proteomic analyses of drought stress responses in sorghum and maize separately, no comparative studies have been performed to identify differences in the molecular events underpinning the greater drought tolerance of sorghum than maize; and therefore, this is the first such cross-species comparison at the proteome level. Bridging this knowledge gap on the adaptive responses to drought across these two species has the potential to enhance drought tolerance in both plant species, since there are differences in the level of drought sensitivity/tolerance within the genotypes of each of the two species. Thus, despite the greater drought tolerance in sorghum than in maize, drought-sensitive sorghum genotypes with various other desirable agronomic traits will benefit if their drought tolerance is improved. A previous study in our laboratory demonstrated the ability of sorghum to sustain growth better than maize under water deficit stress [25]. Furthermore, the study showed that the contrasting responses between the two species were associated with differences in reactive oxygen species (ROS) accumulation and antioxidant enzyme activities [25]. To further understand the molecular events that determine the contrasting responses to water deficit between the two species, a proteomics approach was used to compare the changes in protein expression in the two species under drought. This study measured the relative water content and proline levels in maize and sorghum in response to water deficit and assessed the changes in protein expression of the two plant species under water deficit using label-free quantitative proteomic analysis. This study proposes that the higher level of drought tolerance in sorghum than maize is driven by molecular mechanisms associated with differences in the expression of specific proteins involved in drought responses.

## 2. Materials and Methods

### 2.1. Seed Germination and Plant Growth

The experiment was carried out in a greenhouse at the University of the Western Cape, South Africa (33°55′51.3″ S 18°37′29.2″ E). Sorghum [*Sorghum bicolor* (L.) Moench cv. Superdan] (purchased from Agricol, Brackenfell, South Africa) and maize [*Zea mays* (L.) cv. Borderking] (purchased from McDonalds Seeds, Pietermaritzburg, South Africa) were disinfected with 0.35% (*v*/*v*) sodium hypochlorite for 10 min, followed by 5 rinses using sterile distilled water. Surface sterilized seeds were sown in vermiculite (Windell Hydroponics, Western Cape, South Africa) that had been wetted with 1X nutrient solution [Nitrosol^®^, Fleuron (Pty) Ltd., Johannesburg, South Africa] at room temperature for 3 days to allow their germination. Seedlings were transplanted into cylindrical acrylic tubes (10 cm diameter and a height of 100 cm, with the length covered in foil) containing 7.8 L of Promix Organic (Windell Hydroponics, South Africa), which was saturated with 1% fertilizer (*v*/*v*) [Nitrosol^®^, Fleuron (Pty) Ltd., Johannesburg, South Africa]. The plants were grown at 29 °C during the day (16 h light) and at 19 °C during the night (8 h dark), with a photon flux density reaching 400 µmol m^−2^ s^−1^ during the day phase. Plants received 500 mL of tap water every second day until the V1 stage of growth.

### 2.2. Water Deficit Treatment

Two sets of plants, namely well-watered plants (WW, for which there were ten maize plants and ten sorghum plants) and others subjected to water deficit (WD, for which there were ten maize plants and ten sorghum plants) were used in this study. The WW (control) plants were irrigated with water (500 mL) at intervals of two days until they were harvested (V8 stage of vegetative growth). Five WW maize plants and five WW sorghum plants that showed uniform growth (height, leaf number and morphological appearance) within each of the two species were selected for further analyses. To induce water deficit that simulates drought, WD plants were provided with 100 mL of water (20% of the water supplied to WW plants) once a week, which was stopped when plants reached V3 stage of growth. The WD plants were grown henceforth without further water supply until they showed signs of drought stress (three to four of the oldest leaves turned brown). This corresponded to 40 days after complete water withholding for maize and 55 days after water withholding for sorghum. At these points of water deficit treatment, only five WD maize plants and only five WD sorghum plants exhibiting uniformity in growth within the species were selected for further analyses. The four youngest leaves from maize and sorghum, which were still green, were harvested from each of the selected WW and WD plants. At the time of harvest, maize and sorghum plants were all green and looked healthy, except for the two oldest leaves (at the bottom) in plants grown under water deficit, for which these two leaves were turning dry and started browning, and additionally the four youngest leaves of maize (but not sorghum) showed visible leaf rolling despite still being green and looking healthy. A sample of the Promix Organic growth medium was taken at a depth of 30 cm from the surface of the medium and used to measure the water potential of the growth medium on a WP4C Water PotentioMeter (Meter Group, Pullman, WA, USA) to assess the water status of the soil at the time of harvest, since this is essential in interpretation of the responses of the plants to water deficit. The harvested plant material was rapidly frozen in liquid N_2_, ground into a fine powder and stored at −80 °C until further processing.

### 2.3. Relative Water Content

The youngest fully expanded leaf from each of the selected five plants were used to measure leaf relative water content (RWC). Segments (10 cm long) from the tip of each leaf were obtained and their fresh weights were determined by weighing the segments on a fine weighing balance. The leaf segments were incubated for 4 h in Petri dishes filled with distilled water under ambient light. The turgid weight was measured after blotting off the excess water on the leaf surface. Thereafter, the leaves were dried in an oven at 60 °C for 72 h, after which their dry weights were immediately recorded. The formula RWC = [(FW − DW) ÷ (TW − DW)] × 100 was used for calculation of the RWC, where FW is the fresh weight, DW is the dry weight and TW is the turgid weight.

### 2.4. Proline Content

Proline content was measured from frozen tissue of the four youngest fully expanded leaves from each of the selected five plants, based on a microplate method for small tissue amounts [26]. For these measurements, plant tissue (100 mg) was mixed with 500 μL of sulfosalicylic acid (3% C_7_H_6_O_6_S) and centrifuged for 5 min at 13,000× *g* at room temperature. In a 2 mL microcentrifuge tube, the supernatant (100 μL) was mixed with a reaction mixture (500 μL) consisting of 20% (*v*/*v*) of the 3% sulphosalicylic acid extract (i.e., 100 μL), 40% (*v*/*v*) glacial acetic acid (CH_3_COOH) and 40% (*v*/*v*) acidic ninhydrin (C_9_H_6_O_4_). After mixing, the reaction solution was incubated for 60 min at 100 °C. After cooling in ice for 5 min, the reaction solution was mixed with 99.9% toluene (1 mL of C_6_H_5_CH_3_) and incubated at room temperature for 5 min. Absorbance of the solution at 520 nm was measured using a POLARstar Omega multimode microplate reader (BMG Labtech, Offenburg, Germany). A standard curve was prepared with L-proline and used to determine proline content [26].

### 2.5. Protein Extraction

A modified SDS/phenol extraction method previously described by Wang et al. [27] was used for total soluble protein extraction. The experiment consisted of five independent biological replicates of each species under well-watered and water deficit conditions. Leaf tissue (1 g) was added to 0.5 g polyvinylpolypyrrolidone (PVPP) in a pre-cooled mortar and ground into fine powder with a pestle in liquid nitrogen. The powder was homogenized with 2 mL of 10% TCA/acetone (*w*/*v*) and split equally between two microcentrifuge tubes (one for SDS-PAGE gel analysis and one for label-free liquid chromatography/mass spectrometry analysis). The homogenate was centrifuged for 20 min at 13,000× *g* at 4 °C. This was followed by washing of the pellet twice with pre-cooled ammonium acetate (1 M) in methanol (80% *v*/*v*) and three times with pre-cooled 80% (*v*/*v*) acetone. The supernatant was discarded after each wash. After air drying, the pellet was dissolved in 0.5 mL of buffer containing 2% (*w*/*v*) sodium dodecyl sulphate (SDS), Tris-HCl (0.1 M, pH 8.0), phenylmethylsulfonyl fluoride (PMSF) at a final concentration of 1 mM, 5% (*v*/*v*) β-mercaptoethanol (BME) and 30% (*w*/*v*) sucrose (Sigma, St. Louis, MI, USA). The suspension was mixed with 0.5 mL of phenol (Tris-buffered, pH 8.0) and centrifuged at 4 °C for 20 min at 13,000× *g*. The phenolic layer was taken and mixed with cold 80% (*v*/*v*) methanol, which contained ammonium acetate at a final concentration of 0.1 M. The samples were incubated at 4 °C overnight to precipitate the extracted proteins. The mixture was centrifuged for 20 min at 4 °C at 13,000× *g*. The pellet was washed with cold ammonium acetate (0.1 M, prepared in methanol), followed by a second wash with cold acetone (80% *v*/*v*). After removal of the acetone, the pellet was vacuum-dried in a desiccator at room temperature. The protein pellet for 1-D SDS-PAGE was solubilized in 100 μL of solubilization buffer made up of 2 M thiourea, 4% (*w*/*v*) 3-[(3-cholamidopropyl) dimethylammonio]-1 propanesulfonate (CHAPS), 7 M urea and dithiothreitol (DTT) at a final concentration of 20 mM. The second set of pellets was used for the proteomic analysis. The concentration of solubilized proteins was determined using the Bradford method [28]. The quality of the extracted proteins was assessed using electrophoresis on 1-D SDS-PAGE.

### 2.6. Preparation of Protein Samples for LC–MS/MS Analysis

#### 2.6.1. Solubilization and Quantification of Proteins

Protein pellets from above were resuspended in solubilization buffer [50 mM triethylammonium bicarbonate (TEAB), 2% SDS] and incubated for 5 min at 95 °C. Solubilized proteins were clarified by centrifugation for 5 min at 10,000× *g*. Solubilized proteins were quantified using the QuantiPro BCA assay kit as described by the manufacturer (Sigma).

#### 2.6.2. On-Bead Protein Digestion and HILIC Enrichment

Magnetic beads for hydrophilic interaction liquid chromatography (HILIC) were rinsed twice, for 1 min each time, with 250 μL of washing solution consisting of 15% acetonitrile (ACN) and ammonium acetate (100 mM) at pH 4.5. The beads were dissolved in a loading buffer containing 30% ACN and ammonium acetate (200 mM) at pH 4.5. All of the subsequent steps described hereafter were carried out using a Hamilton MassSTAR robotic liquid handling system (Hamilton, Switzerland). Protein samples (50 μg each) were added to a protein LoBind plate (Merck, Rahway, NJ, USA). Prior to trypsin digestion, proteins were reduced with 10 mM Tris (2-carboxyethyl) phosphine (TCEP) at 60 °C for 1 h and alkylated with 10 mM methyl methanethiosulphonate (MMTS) for 15 min at room temperature. After reduction and alkylation, HILIC magnetic beads were added to the samples in an equivalent volume and incubated for 30 min on a plate shaker at room temperature at 900 rpm. The beads were washed twice with 500 μL of wash buffer (95% ACN) for 1 min each before trypsin digestion. Trypsin (Promega, Madison, WI, USA) was added at a 1:10 ratio (trypsin:protein), followed by incubation on a shaker at 37 °C at 900 rpm for 4 h. The resulting peptides were collected and dried under vacuum, followed by resuspension in 0.1% trifluoroacetic acid (TFA) and desalted. The desalted digests were vacuum-dried once again and subsequently resuspended in loading buffer (2.5% ACN, 0.1% formic acid (FA) prior to analyses.

#### 2.6.3. Liquid Chromatography–Tandem Mass Spectrometry (LC–MS/MS)

Peptides were subjected to LC–MS/MS analyses on a Q-Exactive quadrupole Orbitrap mass spectrometer (Thermo Fisher Scientific, Waltham, MA, USA), which was directly coupled to a nano-HPLC system (Dionex Ultimate 3000). The peptides were dissolved in 2% ACN and 0.1% formic acid (Sigma) and injected into a column (C18 trap) in 3.5% solvent B (0.1% FA, 0.1% ACN) at a flow rate of 5 μL/min for 4 min. Peptides were chromatographically separated on a C18 column (PepAcclaim). Peptides were eluted using a multi-step LC gradient generated at 300 nL/min flow rate as follows: 3.5–9% Solvent B over 6 min, 9–24.6% Solvent B over 45.5 min, 24.6–38.7% Solvent B over 2 min, 38.7–52.8% Solvent B over 2.1 min and 52.8–85.4% solvent B over 0.4 min. The gradient was held at 85.4% solvent B for 10 min, returned to the starting condition (3.5% solvent B), which was held for 15 min. The mass spectrometry system was performed with the capillary temperature set at 320 °C on positive ion mode (at +1.95 kV electrospray). Details of data acquisition on the Q Exactive quadrupole Orbitrap mass spectrometer, which was fitted with a higher-energy collisional dissociation (HCD) cell, are shown in the Appendix A.

### 2.7. Bioinformatics

#### 2.7.1. Data Source

In this experiment, a total of 20 LC–MS/MS runs (81 min runs) were conducted on the Thermo Q-Exactive mass spectrometer. Three extra LC–MS/MS runs represented pools of all samples. A total of 30,000 MS/MS spectra were obtained for each plant species. Protein sequence databases downloaded from the Phytozome protein database on 11 October 2017, held 88,760 proteins for maize and 47,121 proteins for sorghum.

#### 2.7.2. Peptide and Protein Identification Pipeline

Raw data files (spectra) were converted into mzML using the ProteoWizard 3.0 msConvert tool [29]. Peak-Picking and Zlib compression were employed. Database searching employed the MS-GF+ search engine [30] to identify the potential peptides shaped by semi-tryptic specificity, and a 20 ppm precursor tolerance was applied. The data retrieval results were refined by IDPicker (version 3.1) [31] to produce a 2% peptide–spectrum match (PSM) false discovery rate (FDR), with two unique peptides required for each protein.

NCBI BLAST 2.5.0+ makeblastdb [32] was used to index the FASTA sequence databases for ortholog identification between the two species. The blastp program was used to query each sorghum sequence in the maize database and to query each maize sequence in the sorghum database. The generated ortholog files were read in R statistics script and a minimum bit score of 50 was applied. Matches that exceed this threshold were considered true. In cases where multiple matches were found, only the hit with the highest bit score value was retained. Ortholog data and spectral counting tables from IDPicker were read in a script in the R statistical environment to align the spectral count row for each sorghum protein with the spectral count row for the orthologous maize protein. When maize and sorghum orthologs were split into different rows (for example, in the case of paralogs for one species but not the other), the two rows were merged to form one joint row. Proteins lacking orthologs or those with unidentified orthologs were not subjected to further analyses.

#### 2.7.3. Statistical Analysis

Five biologically replicated comparisons of contrasting cohorts (control and water deficit) were used, with target-decoy searching employed to limit aggregate PSM error to 1%. The spectral count data were compared in an R statistics script, with a minimum information criterion of 10 spectra per protein being set, after which a Quasi-Poisson regression was conducted with treatment (well-watered/water deficit) and species variables. Values for multiple testing were corrected via the Benjamini–Hochberg FDR method [33]. If a protein had a q-value < 0.05, it was considered significantly different, with 5% of the claimed changes expected to be false positives.

One-way analysis of variance (ANOVA) was used to analyze the physiological and biochemical results and their significance was determined using the Tukey–Kramer method at a 5% significance level.

#### 2.7.4. Gene Ontology and KEGG Analysis

Differentially expressed proteins were functionally annotated using the Blast2GO program implemented in the OmicsBox v2.2.4 software [34]. The sequence information of the differentially expressed orthologs were obtained from the UniProtKB website in FASTA format on 19 September 2022. Protein sequences were searched, using the basic local alignment search tool (BLAST), against sequences in NCBI (National Center for Biotechnology Information) via the BlastP search algorithm to determine similarity matches. The BLAST search was carried out using the default parameters with a maximum of 20 hits, at an expectation value of 1.0 × 10^−3^, with 33 as the high-scoring segment pair (HSP) length cutoff and 0 as the HSP-hit coverage, with application of a low complexity filter. Sequences that received the best BLAST hits were mapped and annotated using default settings (annotation cutoff 55, Go weight 5, e-value 1.0 × 10^−6^, HSP-hit coverage cutoff 0 and hit filter 500). Proteins were annotated according to gene ontology (GO) by ‘level 2’ on the basis of molecular function (MF), biological process (BP) and cellular component (CC). Protein sequences were scanned for conserved domains against signatures in InterPro using the InterProScan tool, which was an inbuilt program of Blast2GO. Annotated sequences were linked to metabolic pathways via Enzyme Commission (EC) numbers using the KEGG (Kyoto Encyclopedia of Genes and Genomes) extension of Blast2GO.

## 3. Results

### 3.1. Drought Induced the Accumulation of Free Proline in Maize and Sorghum

At the time of harvesting, the water potential of the soil in which the plants were grown under well-watered conditions was −0.12 MPa for the soil used to grow maize and −0.14 MPa for the soil used to grow sorghum, which was not statistically different between the two sets of soil (Figure 1a). This was different for the soil water potential at the time of harvesting of the plants grown under water deficit, where the soil water potential in the maize experiments was −0.75 MPa and it was a statistically different value from the −0.98 MPa obtained from the soil used in the sorghum experiment (Figure 1a). The exposure to drought stress significantly influenced the physiological and biochemical traits of both plant species, as reflected by the decreased relative water content in maize under the water deficit treatment (Figure 1b). In response to water deficit, maize showed a sharp decrease in relative water content (30%), whereas the relative water content in sorghum leaves did not significantly decrease, as shown in Figure 1b. Free proline content was significantly higher in maize leaves (an increase by 60%) than in sorghum leaves (increase was limited to 50%), as depicted in Figure 1c. Interestingly, under drought conditions, the accumulation of free proline in sorghum roots increased by 60% whereas it increased only by 40% in maize roots (Figure 1d).

### 3.2. Differentially Regulated Proteins between Sorghum and Maize

An initial quality control via SDS-PAGE established that protein degradation was minimal (Appendix A). The LC–MS/MS analysis yielded 3154 distinct peptides, including 2752 entries for maize and 2794 for sorghum. There were no matches for 718 (23%) peptides in both plant species for which an ortholog had been named. Proteins named in the ortholog map were constituted from 945 peptides (30%). Furthermore, 416 proteins (13%) consisted of orthologs of both plant species. Of the identified peptides, there were 1070 entries (34%) that consisted of a protein for which another orthologous protein was identified. These 1070 peptides formed 535 rows, which joined the maize spectra from one of the composite rows and the sorghum spectra from a different row. This improved the number of protein sequences for which the sorghum profile and maize profile could be matched, from 416 to 951 distinct proteins (+129% improvement). The Quasi-Poisson model (based on a cut-off of at least 10 spectral counts with a q-value less than 0.05) revealed that 207 orthologs differed in abundance between the two species, irrespective of the treatment (Appendix A). Among the 207 orthologs, 4 proteins (Table 1) were differentially expressed between maize and sorghum in response to drought. These four differentially expressed orthologs thus define the different responses of the two species under water deficit stress. Therefore, the relative differential expression of these proteins between maize and sorghum as well as their functional fates in response to drought stress are further addressed herein.

### 3.3. Gene Ontology and KEGG Pathway Annotation

Differentially expressed orthologs between maize and sorghum (Table 1) were characterized according to Gene Ontology (GO). As shown in Figure 2, GO enrichment analysis revealed that the metabolic process and cellular process were the most represented biological processes, followed by the response to stimulus and biological regulation. Within the molecular function category, catalytic activity and binding were the most enriched. According to the cellular component GO terms, the differentially expressed proteins were mainly localized in the cellular anatomical entity.

To better understand the functions of the differentially regulated proteins between maize and sorghum, orthologs were assigned to different metabolic pathways via the Kyoto Encyclopedia of Genes and Genomes (KEGG) pathway database. The KEGG pathway analyses showed that the differentially expressed proteins were associated with the biosynthesis of phenolic acids, biosynthesis of indole acetic acid, sucrose and D-fructose metabolism, ROS scavenging and biosynthesis of melanin-related compounds (Figure 3).

## 4. Discussion

### 4.1. Proline Accumulation in Sorghum Roots Was Associated with Improved Water Retention

The reduction of RWC in maize leaves, which did not occur in sorghum, indicated better water retention ability in sorghum under drought than in maize. A similar observation was reported by Hasan et al. (2017) [35], where drought stress significantly decreased the leaf RWC in maize, while no significant effect was observed in sorghum. Compatible solutes act as osmoprotectants and mediate osmotic adjustment in plants under water deficit [36]. Among them, free proline is the most common osmolyte occurring in plants grown under water deficit [37]. Therefore, high accumulation of proline can enhance water retention capacity [38]. In this study, water deficit increased the proline content in the roots and leaves of both species. However, compared to maize, sorghum demonstrated a greater increase in proline content in the root. Such enhanced proline accumulation in the sorghum roots would result in a higher degree of decrease in water potential in sorghum roots than in maize roots, which would allow for better water uptake from the soil by sorghum roots than maize roots, and hence sustain shoot water status longer in sorghum than maize.

### 4.2. Gene Ontology and KEGG Pathways

A combination of gene ontology and Kyoto Encyclopedia of Genes and Genomes analyses can link physiological changes to molecular pathways, which can facilitate the identification of the pathways mediating the effects of environmental stresses in the plant [39]. In this study, some differentially expressed proteins between maize and sorghum were involved in various cellular process, metabolic process, catalytic activity and stimulus response. Proteins with catalytic function act as pivotal regulators involved in multiple processes of plant development and responses to environmental changes, through modulation of downstream protein activities [40]. Exploring the function of the enzymes and their associated pathways could provide deeper insight into the mechanisms underlying sorghum adaptation to drought stress. To gain this understanding, the pathways with potential roles in plant stress responses are further discussed. Phenylalanine/tyrosine ammonia-lyase (EC 4.3.1.25) plays a key role in the phenylpropanoid biosynthesis pathway (ko00940, Table 2 and Figure 3) and was downregulated in both maize and sorghum (0.9- and 1.0-fold, respectively). Three enzymatic activities are central to the phenylpropanoid biosynthesis pathway. This includes the non-oxidative elimination of ammonia from L-phenylalanine and L-tyrosine by phenylalanine/tyrosine ammonia-lyases (PAL/PTALs) to produce trans-cinnamic acid and p-coumaric acid, respectively. In the second step, cinnamic acid 4-hydroxylase (C4H) catalyzes the hydroxylation of trans-cinnamic acid to 4-coumarate.

Lastly, 4-coumarate is activated by 4-coumarate-CoA ligase (4CL) to form 4-coumaroyl-CoA [41]. The p-Coumaroyl-CoA enters different downstream pathways, which leads to the biosynthesis of numerous compounds with antioxidant properties, including monolignol, coumarin, stilbene and flavonoids. In a previous study, salt stress increased the expression of PTAL in *Zea mays* [42]. These reports contradicted the results observed both in our study and another preceding study [43] on stressed *Medicago sativa* L., where a decreased abundance of PAL was correlated with elevated levels of cinnamic acid. Our recent assessment of the maize response to water deficit suggested that drought leads to altered levels of phenolic acids, driven by changes in the expression of genes encoding cinnamate 4-hydroxylase and p-coumaric acid 3-hydroxylase [44]. Thus, assessing the levels of phenolic acids and flavonoids in sorghum and maize will contribute to the understanding of how enzymes in the phenylpropanoid biosynthesis pathway influence the responses to drought in these two C4 plant species.

Indole-3-acetaldehyde oxidase (IAA oxidase, EC 1.2.3.7), which is part of the tryptophan biosynthetic pathway (ko00380, Table 2 and Figure 3), was over-expressed in both species. Although *Zea mays* showed a greater fold change (2.6) than *Sorghum bicolor* (1.1) in response to water deficit, indole-3-acetaldehyde oxidase abundance was higher in sorghum than in maize irrespective of whether the plants were grown in well-watered or water deficit conditions (Appendix A). Therefore, it can be proposed that the higher expression of indole-3-acetaldehyde oxidase in sorghum could lead to greater production of metabolites associated with the tryptophan biosynthetic pathway. Hence, this would mean that the greater abundance of this enzyme in sorghum than in maize translates to higher products of this pathway in sorghum compared to maize. In the tryptophan biosynthesis pathway, indole-3-acetaldehyde (IAAl) is oxidized by IAAl oxidase to produce indole-3-acetic acid (IAA) [45]. Indole-3-acetic acid is the most commonly occurring auxin in plants. Auxin is a key hormone that plays vital roles in plant growth and development, which include cell division, cell differentiation and cell elongation [46]. As a regulator, auxin mediates the signaling pathways in plant responses to stress [47]. Given that indole-3-acetaldehyde oxidase produces indole-3-acetatic acid, we suggest that sorghum tolerance to drought is mediated by the higher accumulation of indole-3-acetate in sorghum upon water deficit. This is based on compelling evidence showing that high levels of indole-3-acetate lead to drought tolerance [48]. Such indole-3-acetate-mediated drought tolerance occurs via the activation of genes related to auxin, abscisic acid and jasmonic acid biosynthesis [48].

Sucrose synthase (EC 2.4.1.13) was linked to the starch and sucrose metabolism pathway (ko00500, Table 2 and Figure 3). Sucrose synthase showed a decrease in abundance (1.5- and 2.3-fold) in maize and sorghum, respectively. In plants, sucrose synthase is involved in the hydrolysis of sucrose, leading to the production of UDP-glucose and D-fructose (or ADP-glucose) [49]. A recent study demonstrated that the activity of sucrose synthase was decreased in sorghum when grown under osmotic stress [50]. As organic osmolytes, sucrose or D-fructose have an important role in regulating the osmotic gradient in cells to maintain water status in plants [51]. Interestingly, the decrease in sucrose synthase expression under drought was higher in sorghum than in maize. The higher reduction in sucrose synthase expression in sorghum may be linked to the greater water retention capacity in sorghum than maize under water deficit, thus implying that sucrose synthase activity is only required in cases where water deficit stress is experienced in the plant to necessitate osmotic adjustment through sucrose or D-fructose.

Polyphenol oxidase I (EC 1.10.3.1), which catalyzes the initial reactions in the tyrosine metabolism pathway (ko00350, Table 2 and Figure 3), was upregulated in both plant species in response to water deficit stress. Polyphenol oxidases possess catechol oxidase activity. Even though the expression of polyphenol oxidase I/catechol oxidase increased in both maize and sorghum by 3.0- and 1.9-fold, respectively, it was considerably higher in sorghum under both water treatments (WW and WD). Therefore, its metabolic products are likely more in sorghum than in maize under both water status conditions. Catechol oxidase can regulate the biosynthesis of melanins and other polyphenolic compounds by catalyzing the oxidation of DOPA to DOPA–quinone [52]. The adaptive role of catechol oxidase during plant exposure to drought is not yet well known. However, the evidence reporting that hydrogen peroxide is utilized as a cofactor in the oxidation of DOPA and dopamine during the process of melanogenesis has been presented [53]. These observations were supported by research demonstrating that catechol oxidase has the catalytic activity of catalase [54]. According to these authors, two catechol oxidase isoforms (39 kDa and 40 kDa) from sweet potato (*Ipomoea batatas*) were tested for catalase activity by applying H_2_O_2_ as a substrate. Their results showed that the 39-kDa protein exhibits catalase enzymatic activity, but not the 40-kDa protein. Furthermore, it was proposed that the catalytic mechanism is based on the binding of two molecules of hydrogen peroxide to the active site of the enzyme [54]. Therefore, catechol oxidase can act as a ROS scavenger by detoxifying hydrogen peroxide into O_2_ and H_2_O, as catalase does, and/or impart plant stress tolerance through the production of phenolic compounds, which regulate important defense mechanisms in plants against water deficit stress. Furthermore, given that catechol oxidase is a phenol oxidase and the increased activity of phenol oxidase is associated with improved drought tolerance [55], the enhanced drought tolerance in sorghum can be attributed partly to the more pronounced abundance of catechol oxidase observed in sorghum than in maize. Therefore, the greater abundance of catechol oxidase in sorghum under both water conditions possibly contributes to the better ability of sorghum to withstand water deficit than maize.

## 5. Conclusions

In this study, drought stress reduced the RWC of maize leaves but not sorghum leaves. In addition, *Sorghum bicolor* displayed a considerable increase in free proline content in roots and showed better capability to maintain water status than *Zea mays*. This supports the notion that *Sorghum bicolor* withstands water stress better than *Zea mays*. Importantly, the leaf proteome profiling revealed different response patterns in these two cereal crops. Furthermore, our findings indicate that the better drought tolerance of sorghum than maize involved the regulation of some enzymes, with PTAL, sucrose synthase, indole-3-acetaldehyde oxidase and catechol oxidase being among these enzymes. Proteins with PTAL activity are required for the synthesis of cinnamic acid and p-coumaric acid, and the observed changes in PTAL abundance implied a role of phenolic acids in drought tolerance. As an osmolyte, sucrose plays an important role in plant osmotic regulation, enabling sorghum to retain water better than maize. In short, the higher decrease in sucrose synthase expression in sorghum is possibly related to its ability to maintain water status better than maize under drought. The differential water deficit-induced expression of indole-3-acetaldehyde oxidase may positively contribute towards the growth of sorghum despite the water limitation. Alterations in catechol oxidase, which has catalase activity, could also contribute to efficient scavenging of stress-induced ROS in sorghum compared to maize, and this may involve downstream products of the phenol oxidase-like activity in the catechol oxidase. This study thus identified proteins whose encoding genes could be targeted for the improvement of maize and sorghum tolerance to drought, as represented in the schematic proposed for conferring drought tolerance (Figure 4).

Such drought tolerance can be achieved through marker assisted selection to select varieties of maize and sorghum with expression profiles of these genes that follow patterns of expression in drought-tolerant genotypes of sorghum, or through altering the expression of these genes in maize and sorghum through genetic engineering to achieve similar patterns of their expression as in drought-tolerant sorghum genotypes. Such biotechnological approaches are important for sustaining maize and sorghum production during drought, which will contribute positively to food security. This is because these crops are critical for food security in Africa and globally, based on their extensive use as food for humans and feed for animals, in addition to their industrial uses (mainly as starch and biofuel). The use of only one sorghum and only one maize genotype in this study limits the number of proteins that can be associated with drought responses in the two species. This limitation is also prohibitive in concluding whether the changes observed in the water deficit-induced differences in protein expression between sorghum and maize are associated with drought tolerance or drought sensitivity. To resolve this limitation, future work will involve the screening of several genetically diverse genotypes of sorghum and maize to include a number of drought-sensitive and drought-tolerant genotypes of both species and subject these diverse genotypes to similar proteomic analysis. This will allow for the identification of regulated proteins based on whether such proteins are upregulated or downregulated in the drought-sensitive or the drought-tolerant genotypes, and thus enable us to distinguish between proteins associated with tolerance from those associated with sensitivity to drought. Despite these limitations, this study clearly shows which subset of proteins and pathways are important in distinguishing the responses of maize from those of sorghum in water deficit conditions.

## Figures and Tables

**Figure 1 life-13-00170-f001:**
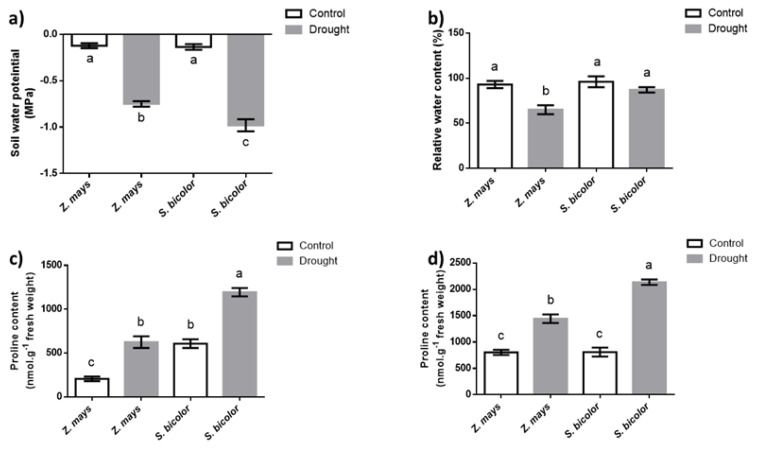
Changes in soil water potential (**a**), relative water content (**b**), and free proline accumulation in leaves (**c**) and roots (**d**) of maize and sorghum under water deficit. Data presented are means (±SE) of five independent experiments (n = 5). Bars with different letters are significantly different at *p* ≤ 0.05.

**Figure 2 life-13-00170-f002:**
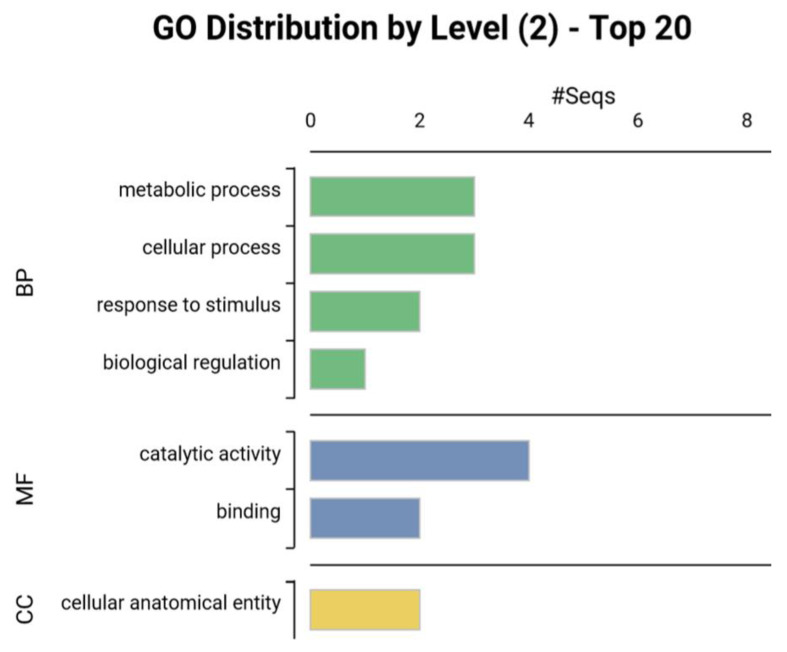
Gene ontology (GO) analysis of proteins differentially regulated between maize and sorghum under water deficit stress as determined by Blast2GO according to GO distribution by level 2. Molecular function (MF); cellular component (CC); biological process (BP).

**Figure 3 life-13-00170-f003:**
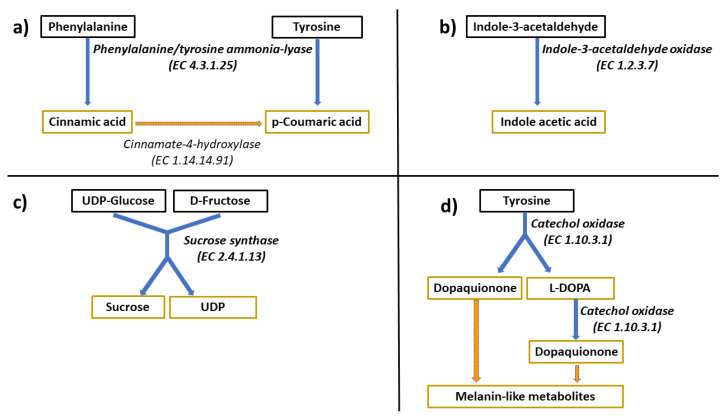
KEGG pathway analysis revealing proteins differentially regulated between maize and sorghum under water deficit stress. Phenylalanine and tyrosine metabolism affecting phenolic biosynthesis via phenylalanine/tyrosine ammonia lyase (PTAL) is illustrated (**a**), along with conversion of indole-3-acetyldehyde to indole acetic acid by indole-3-acetaldehyde oxidase (**b**), production of sucrose/D-fructose from UDP glucose via a reaction catalyzed by sucrose synthase (**c**) and production of DOPA quinone via a reaction catalyzed by catechol oxidase to metabolize tyrosine towards synthesis of melanin-like compounds (**d**). Metabolites encased in black rectangles are substrates for the enzymes (described by italicized bold font, with indicated Enzyme Commission numbers). Metabolites enclosed in gold rectangles are products of the indicated enzymes. The gold dashed arrow or its associated enzyme (in italicized regular font, not bold) indicates a section of a metabolic pathway for which the catalyzing enzyme was not identified as differentially regulated between maize and sorghum but for which the products of the identified differentially regulated proteins are substrates for downstream proteins in the pathway.

**Figure 4 life-13-00170-f004:**
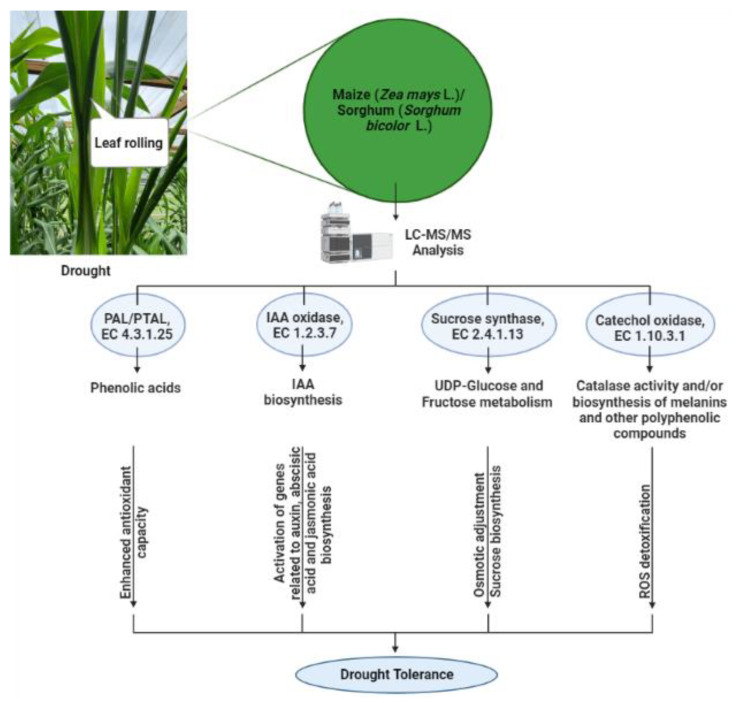
Proposed mechanism through which proteins differentially regulated between maize and sorghum under water deficit stress lead to drought tolerance. Drought-induced phenylalanine and tyrosine metabolism leading to phenolic acid biosynthesis via phenylalanine/tyrosine ammonia lyase (PAL/PTAL, i.e., PTAL) enhances antioxidant activity. When activation of indole acetic acid oxidase occurs, auxin biosynthesis is enhanced and can act coordinately with the biosynthesis of abscisic acid and jasmonic acid to regulate plant responses to water deficit, leading to drought tolerance. Furthermore, improved drought tolerance can be achieved by regulation of sucrose synthase to enhance osmotic adjustment through sucrose and D-fructose metabolism, and through catechol oxidase-mediated detoxification of ROS that can be coupled to biosynthesis of melanin-related and other phenolic compounds. BioRender (biorender.com) was used to create the figure.

**Table 1 life-13-00170-t001:** Protein orthologs with differential expression between sorghum and maize in response to water deficit stress.

Accession	Description	Log2 Fold Change		Q-Value Species	Q-Value Treatment
		Maize	Sorghum		
GRMZM2G141473_P01/ Sobic.001G062300.1.p	Indole-3-acetaldhyde oxidase	2.6	1.1	0.000187049	0.044375583
GRMZM2G319062_P01/ Sobic.007G068500.1.p	polyphenol oxidase I, chloroplastic-like	3.0	1.9	0.000564461	0.009718389
GRMZM2G074604_P01/ Sobic.004G220300.1.p	phenylalanine/tyrosine ammonia-lyase	−0.9	−1.0	0.000403801	0.00363421
GRMZM2G152908_P01/ Sobic.001G344500.2.p	sucrose synthase 2	−1.5	−2.3	0.000451047	0.000119933

A negative sign indicates a decrease in protein expression. Maize proteins are depicted in the top accession number (starting with GRMZM) and sorghum proteins are depicted in the bottom accession number (starting with Sobic).

**Table 2 life-13-00170-t002:** Functions of the enzymes differentially expressed between *Zea mays* and *Sorghum bicolor* in response to drought.

Enzyme Code (EC)	Name of Enzyme	Sequences	Substrates	Products
1.2.3.7	Indole-3-acetaldhyde oxidase	GRMZM2G141473_P01/	Indole-3-acetaldehyde	Indole acetic acid
		Sobic.001G062300.1.p		
1.10.3.1	Polyphenol oxidase I, Catechol oxidase	GRMZM2G319062_P01/	Tyrosine	L-DOPA
		Sobic.007G068500.1.p		Dopaquionone
4.3.1.25	Phenylalanine/Tyrosine ammonia-lyase	GRMZM2G074604_P01/	Phenylalanine	Cinnamic acid
		Sobic.004G220300.1.p	Tyrosine	p-Coumaric acid
2.4.1.13	Sucrose synthase 2	GRMZM2G152908_P01/	UDP-Glucose	Sucrose
		Sobic.001G344500.2.p	D-Fructose	UDP

EC represents the Enzyme Commission number. Maize proteins are depicted in the top accession number (starting with GRMZM) and sorghum proteins are depicted in the bottom accession number (starting with Sobic). UDP is uridine diphosphate, L-DOPA is L-3,4-dihydroxyphenylalanine.

## Data Availability

All data are available upon request in addition to data provided in the Appendix A.

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
