# Peer review of "Comparative Proteomics Analysis between Maize and Sorghum Uncovers Important Proteins and Metabolic Pathways Mediating Drought Tolerance"

_life, 2023, doi:10.3390/life13010170_

Round 1
Reviewer 1 Report
In this manuscript titled “Comparative Proteomics Analysis between Maize and Sorghum Uncovers Important Proteins and Metabolic Pathways Mediating Drought Tolerance” by Ali et al. doing a comparative study to determine the drought tolerance related metabolic pathways in maize and sorghum. Study aims to identify molecular differences between maize and sorghum that are related to drought stress tolerance in sorghum. Authors analyzed protein expression in the two plant species under drought to identify drought stress responsive proteins. They identify important molecular targets which would contribute towards development of genetically improved maize and sorghum with superior tolerance to drought. Overall study was good and will help the researcher to understand the metabolic pathways and linked genes with drought stress tolerance in crops. But need to address some comments:
Some major and minor suggestions are below
- The structure of the abstract is not clear. Please re-write. Add your core findings.
- Please enhance the figures quality most of the them not readable.
- The authors need to mention complete tools (bioinformatics) names along with abbreviation where applicable.
- Update the introduction and discussion part with more related information. Add most recent literature that are directly linked with your findings
- Some of the references are too old, please update the references with new one where required.
- There are some ambiguities in Table 2-5. Either its Table or Figures. Please clarify.
- English language, and grammar need more attention. Most of the sentences are not much clear.
Author Response
Thank you for the useful input you have made to our manuscript.
We would like to point out the following, based on your input:
1. The English has been improved using a native English speaker.
2. The introduction has been improved as recommended.
3. We have provided more detail on the experimental design and improved the description of the methodology as recommended from your comments.
On specific queries, we provide the following responses:
Question/Comment 1: The structure of the abstract is not clear. Please re-write. Add your core findings.
Answer 1: We have re-written the abstract to make it clearer and we added our core findings to the abstract as recommended.
Question/Comment 2: Please enhance the figures quality most of the them not readable.
Answer 2: We have enhanced the quality of the figures and all of them are now easily readable.
Question/Comment 3: The authors need to mention complete tools (bioinformatics) names along with abbreviation where applicable.
Answer 3: We have mentioned and described complete bioinformatics tool names and provided abbreviations as recommended.
Question/Comment 4: Update the introduction and discussion part with more related information. Add most recent literature that are directly linked with your findings
Answer 4: We have updated the introduction and discussion parts as requested and added most recent literature as requested.
Question/Comment 5: Some of the references are too old, please update the references with new one where required.
Answer 5: We have provided newer references as requested.
Question/Comment 6: There are some ambiguities in Table 2-5. Either its Table or Figures. Please clarify.
Answer 6: We have cleared the ambiguities by separating figures from tables and merging some information in some of the tables into one table.
Question/Comment 6: English language, and grammar need more attention. Most of the sentences are not much clear.
Answer 6: We have improved the English language and grammar by using the service of a native English speaker and made the sentences clearer.
Reviewer 2 Report
The manuscript is well written with very clear methodology and results. The study is interesting and refers to comparative proteomics analyses between maize and sorghum crops for drought tolerance.
My suggestions for the manuscript are:
1) Explain the hypothesis and scientific questions more clearly and objectively in the introduction;
2) Write more clearly and specifically: what is the novelty of the work? There is a sentence in the introduction trying to explain this, but it is still too timid in relation to what was done;
3) Did you use a new methodology in comparison to most other studies? This needs to be highlighted as well.
4) Items 4.1 and 4.2 of the discussion are very superficial. If you created these items in the discussion it is necessary to discuss them in more depth. In item 4.1 for example, you could include discussions such as: are there other species of cultivated plants that are drought resistant and that present these characteristics? Do plants from drier climates also present this characteristic? The climate in which the plant is submitted can change this in some way (naturally)? has this already been reported in any work?
5) You explain that this study could help in food security and in the defence of the plant against drought. I agree and this is very important. But it is necessary to make this relationship of scales clearer in the written text. How, in practice, would these scientific discoveries help? This would make the work much richer and citable in other areas (not only genetics).
Author Response
We take this opportunity to thank you for the valuable comments and suggestion on our manuscript. We have improved the conclusion and given it much more detail that emanates from our results and discussion.
We would like to provide the following responses to your comments:
Q/Comment 1) Explain the hypothesis and scientific questions more clearly and objectively in the introduction.
Answer: We have explained the proposed hypothesis and scientific questions more clearly, indicating the aim and objectives of the study very clearly.
Q/Comment 2) Write more clearly and specifically: what is the novelty of the work? There is a sentence in the introduction trying to explain this, but it is still too timid in relation to what was done,
Answer: We re-written the manuscript more clearly, explained the novelty of the study.
Question/Comment 3) Did you use a new methodology in comparison to most other studies? This needs to be highlighted as well.
Answer: The cross-species comparison was facilitated by a novel approach where ortholog mapping was applied, which is described in detail in the methodology of the revised manuscript.
Question/Comment 4) Items 4.1 and 4.2 of the discussion are very superficial. If you created these items in the discussion it is necessary to discuss them in more depth. In item 4.1 for example, you could include discussions such as: are there other species of cultivated plants that are drought resistant and that present these characteristics? Do plants from drier climates also present this characteristic? The climate in which the plant is submitted can change this in some way (naturally)? has this already been reported in any work?
Answer: We have provided more detail in the discussion on these aspects and included relevant references to cover the instances where similar observations have been made in other plant species.
Question/Comment 5) You explain that this study could help in food security and in the defence of the plant against drought. I agree and this is very important. But it is necessary to make this relationship of scales clearer in the written text. How, in practice, would these scientific discoveries help? This would make the work much richer and citable in other areas (not only genetics).
Answer: We have clarified how these relationships (improvement of drought tolerance and food security) interact and how the described work will contribute towards this goal, with more clearer and expanded text in the discussion and conclusion.
Reviewer 3 Report
Dear Authors,
Your research paper has good findings. However, ı don't suggest putting references in the conclusion part, please delete those, just indicate the importance of your research subject as you mostly did.
Best regards
Author Response
Thank you for the valuable input. With regards to your suggestions, we would like to point our the following.
Suggestion 1: However, ı don't suggest putting references in the conclusion part, please delete those.
Answer 1: We have removed the references in the conclusion.
Suggestion 2: Just indicate the importance of your research subject as you mostly did..
Answer 2: We have indicate the importance of the research work in the conclusion in terms of the contribution of the work to improvement of drought tolerance in crops and the related impact on food security.
Reviewer 4 Report
This article entitled: “Comparative Proteomics Analysis between Maize and Sorghum Uncovers Important Proteins and Metabolic Pathways Mediating Drought Tolerance” investigates the significance of proteomics analysis between maize and sorghum in mitigating drought stress. The methods and experimental design seem reasonable and sound. However, a few suggestions are needed to be incorporated to improve this manuscript's quality. I, therefore, suggest a major revision for this manuscript.
Please delete a simple summary from your manuscript.
Abstract section:
1. Please underscore the scientific value added to your paper in your abstract. Your abstract should clearly state the essence of the problem you are addressing, what you did and what you found and recommend. Moreover, the abstract needs careful reading and should precisely depict important results. It should have a stronger concluding sentence. That will help a prospective reader of the abstract to decide if they wish to read the entire article.
2. Please highlight the key findings of your results in the abstract section with a % difference.
The keywords of this manuscript should be (Proteomics analysis, drought stress; tolerance, maize, sorghum, and drought tolerance)
Introduction section:
1. Lines 46 and 47; please introduce maize and sorghum as world-important cereals crop
2. (Drought stress is considered one of the most significant natural hazards), please highlight drought status in the world, Africa, and South Africa.
3. Explore the impact of drought on plant production, especially in cereals, with clear facts and figures. For example, Farooq et al., (2009) (doi:10.1051/agro:2008021) reported adverse effects of drought in crop production such as Barley (49–57% yield reduction), maize (70–47% yield reduction), rice (30–55% yield reduction) and so on.
4. Please explain the study's novelty, i.e., what is known and what needs to be explored?
5. At the end of the Introduction section, the main objectives of this study should be more detailed and presented
6. What is the hypothesis of the current manuscript?
Material and methods:
1. Please tell the place where the experiment was conducted (greenhouse of any institute or university) (any GPS location of study)
2. What is the condition of plants under water-deficient conditions, any appropriate reference to follow the procedure? If yes, then please mention
3. (its absorbance was measured at 520 nm) line 154, please mention the instrument name, model, manufacturing company name, city, and country.
Results section:
1. Please observe the latest published scientific studies and improve the presentation quality of the results section
(https://doi.org/10.1080/13102818.2020.1805015) and (doi:10.3390/ijms20153743)
2. Figures 3-6 are just basic figures copied from internet sources, adding no novelty in the results section. So, authors are advised to improve the quality of the result section.
Discussion section
1. Overall, the authors generated enough data. However, the presentation of those is not scientifically strong. The entire Discussions section is generally weak and must be strengthened by discussing further.
2. Mention how proteomics analysis influences the drought tolerance mechanism in maize and sorghum under drought stress by showing a schematic diagram.
Conclusions section:
The conclusion is generic and fails to improve the existing knowledge base. The conclusions can still be improved by analyzing where the current work on adsorbents is focused and the remaining gaps in the literature where more research should be conducted. It is recommended to use quantitative reasoning compared with appropriate benchmarks, especially those stemming from previous work. Limitations in the suggested approach should be discussed in the conclusions section. Please add future work as well.
Author Response
We thank the reviewer for the constructive and very useful comments. Our responses are indicated below, which provide descriptions of improvements to the English in the manuscripts, background and references, clearer presentation of the results and the conclusions.
Comment: Please delete a simple summary from your manuscript.
Response: The simple summary is a requirement from the journal, as per journal guidelines.
Comments on the Abstract section:
Comment: Please underscore the scientific value added to your paper in your abstract. Your abstract should clearly state the essence of the problem you are addressing, what you did and what you found and recommend. Moreover, the abstract needs careful reading and should precisely depict important results. It should have a stronger concluding sentence. That will help a prospective reader of the abstract to decide if they wish to read the entire article.
Answer: We have clarified the scientific value in the abstract and re-written the abstract as recommend by the reviewer.
Comment: Please highlight the key findings of your results in the abstract section with a % difference.
Answer: We have indicated how the expression of the proteins differ, in the abstract itself. However, using percentage differences for fold-changes can be misleading and can lead to misinterpretation of the findings, and so we gave more detail of this in the results section itself for clarity.
Comment: The keywords of this manuscript should be (Proteomics analysis, drought stress; tolerance, maize, sorghum, and drought tolerance).
Answer: We have change the keywords as suggested by the reviewer.
Introduction section:
Comment: Lines 46 and 47; please introduce maize and sorghum as world-important cereals crop
Answer: We have done this as suggested by the reviewer.
Comment: (Drought stressis considered one of the most significant natural hazards), please highlight drought status in the world, Africa, and South Africa.
Answer: We have provided the required detail as requested.
Comment: Explore the impact of drought on plant production, especially in cereals, with clear facts and figures. For example, Farooq et al., (2009) (doi:10.1051/agro:2008021) reported adverse effects of drought in crop production such as Barley (49–57% yield reduction), maize (70–47% yield reduction), rice (30–55% yield reduction) and so on.
Answer: We have indicated this with clear facts and figures and included the suggested references.
Comment; Please explain the study's novelty, i.e., what is known and what needs to be explored?
Answer: This has been explained in the revised manuscript in the introduction section and emphasized in the conclusion in relation to the findings on the work.
Comment: At the end of the Introduction section, the main objectives of this study should be more detailed and presented.
Answer: We have improved this by providing more detail of the main objectives.
Comment: What is the hypothesis of the current manuscript?
Answer: The proposed hypothesis of the study has been described in detail in the revised manuscript.
Comments on Material and methods:
Comment: Please tell the place where the experiment was conducted (greenhouse of any institute or university) (any GPS location of study).
Answer: The location and GPS coordinates of the site of study have been provided in the revised manuscript.
Comment: What is the condition of plants under water-deficient conditions, any appropriate reference to follow the procedure? If yes, then please mention.
Answer: The condition of the plants under water deficit has been described in the revised manuscript and the methodology to get them to this condition has been described. No reference is available though, but this is not an issue since the experimental procedure is fully described.
Comment: (its absorbance was measured at 520 nm) line 154, please mention the instrument name, model, manufacturing company name, city, and country.
Answer: We have provided the requested detail in the methodology.
Comments on the Results section:
Comment: Please observe the latest published scientific studies and improve the presentation quality of the results section
(https://doi.org/10.1080/13102818.2020.1805015) and (doi:10.3390/ijms20153743)
Answer: We have improved the presentation quality of the results section and observed the latest published literature on related topics as suggested by the reviewer.
Comment: Figures 3-6 are just basic figures copied from internet sources, adding no novelty in the results section. So, authors are advised to improve the quality of the result section.
Answer: We have improved the presentation of the figures and data in the tables instead of using outputs from the web-based tools.
Comments on the Discussion section
Comment: Overall, the authors generated enough data. However, the presentation of those is not scientifically strong. The entire Discussions section is generally weak and must be strengthened by discussing further.
We have strengthened the discussion section in the revised manuscript by providing more extensive and in-depth discussion of the results obtained.
Comment: Mention how proteomics analysis influences the drought tolerance mechanism in maize and sorghum under drought stress by showing a schematic diagram.
Answer: We have included new diagrams in the manuscript (results and conclusion) and a new table (discussion) to indicate how drought influences drought tolerance mechanism in the two species, linking the identified proteins and their biochemical functions to effects on drought responses.
Comments on the Conclusions section:
Comment: The conclusion is generic and fails to improve the existing knowledge base. The conclusions can still be improved by analyzing where the current work on adsorbents is focused and the remaining gaps in the literature where more research should be conducted. It is recommended to use quantitative reasoning compared with appropriate benchmarks, especially those stemming from previous work. Limitations in the suggested approach should be discussed in the conclusions section. Please add future work as well.
Answer: We have followed this suggestion in detail and the reviewer will ow see that the conclusion is more detailed and expanded, yet very focused, with specific emphasis on the points suggested by the reviewer.
Round 2
Reviewer 4 Report
Now, the article is suitable for publication